# The Impact of Diclofenac Gel on Ion Transport in the Rabbit (*Oryctolagus cuniculus*) Skin: An In Vitro Study

**DOI:** 10.3390/molecules28031332

**Published:** 2023-01-30

**Authors:** Wioletta Dobrzeniecka, Małgorzata Daca, Barbara Nowakowska, Marta Sobiesiak, Karolina Szewczyk-Golec, Alina Woźniak, Iga Hołyńska-Iwan

**Affiliations:** 1Department of Pathobiochemistry and Clinical Chemistry, Faculty of Pharmacy, Ludwik Rydygier Collegium Medicum in Bydgoszcz, Nicolaus Copernicus University in Torun, 87-100 Torun, Poland; 2Department of Inorganic and Analytical Chemistry, Faculty of Pharmacy, Ludwik Rydygier Collegium Medicum in Bydgoszcz, Nicolaus Copernicus University in Torun, 87-100 Torun, Poland; 3Department of Medical Biology and Biochemistry, Faculty of Medicine, Ludwik Rydygier Collegium Medicum in Bydgoszcz, Nicolaus Copernicus University in Torun, 87-100 Torun, Poland

**Keywords:** diclofenac, hypersensitivity, ion transport, pain reaction, transepithelial water transport

## Abstract

Diclofenac belongs to the non-steroidal anti-inflammatory drugs with analgesic, antipyretic and anti-inflammatory effects. Diclofenac administration on the skin may be associated with the appearance of side effects. The study aimed to evaluate the impact of diclofenac gel on transepithelial electrophysiological parameters of the 55 rabbit abdomen skin specimens. The electric parameters were analyzed in a modified Ussing chamber. The resistance (R) of the skin specimens treated with diclofenac gel significantly increased, which could be related to the reduction in the water content in intercellular spaces and, consequently, tighter adhesion of the cells. Increased electric potential (PD) was also observed in the skin specimens treated with diclofenac gel. The increase in both R and PD measured under stationary conditions was most likely caused by a transient and reversible increase in sodium ion transport, as the R and PD values decreased after the diclofenac gel was washed away. However, diclofenac gel did not affect the maximum and minimum PDs measured during stimulations. Therefore, it seems that diclofenac gel does not affect the perception of stimuli in the model system used.

## 1. Introduction

Non-steroidal anti-inflammatory drugs (NSAIDs) are one of the most frequently used medications worldwide due to their availability over the counter and their ability to treat a wide range of pain, inflammation, and/or fever conditions [1,2,3]. Diclofenac (*Diclofenacum natricum*) is a derivative of aminophenylacetic acid that belongs to NSAIDs [3]. Diclofenac inhibits the metabolism of arachidonic acid, preventing the synthesis of thromboxanes and prostaglandins, which generate inflammation, fever, and pain sensation (Figure 1). To be more precise, diclofenac action is based on the inhibition of cyclooxygenases, prostaglandin-endoperoxide synthase (COX, EC 1.14.99.2), including COX-1 and COX-2 isoenzymes [4]. Inhibition of each form leads to different pharmacological effects [3,4]. COX-1 is an enzyme physiologically active in many tissues, including platelets, kidneys, blood vessels, and the stomach. The COX-2 isoform is physiologically active in keratinocytes and involves their maturation and differentiation [4]. Additionally, COX-2 affects the development of hair follicles and hair growth. Moreover, COX-2 is an enzyme whose activity increases rapidly in inflamed tissues [4]. As a drug, diclofenac is characterized by anti-inflammatory, antipyretic, analgesic, and platelet aggregation-inhibiting effects, although in this case, it has a weaker and shorter duration than acetylsalicylic acid [1,2,4,5]. 

Considering its action, diclofenac can be used in degenerative joint disease, long-term inflammations, such as spondylitis, painful menstruation, and after surgery [5,6,7]. Moreover, some studies have shown that, in addition to arthritis and pain, the COX-2 inhibitors such as diclofenac can be used to treat neurodegenerative diseases, including Alzheimer’s disease [8,9], and to reduce the risk of many cancers [10,11,12]. This fast-acting drug can treat actinic keratosis [13] and may have an antituberculosis effect [14]. 

Diclofenac administration may be associated with several side effects. In the case of hypersensitivity to the medicinal substance or any component of a dosage form, a rash, edema, asthma, and hypersensitivity to other NSAIDs may occur (Table 1) [1,2,3]. The form in which the diclofenac is taken significantly affects the rate and amount of the drug absorbed into the subcutaneous tissues. Topical application allows for reducing the toxic effect of the drug significantly. Relatively fast-acting preparations, including gel preparations, are prepared with substances that improve skin penetration [15,16]. The skin’s ability to respond to internal and external stimuli is closely correlated with changes in ion transport, especially sodium and chloride ions [17,18,19,20,21,22]. The transport of sodium ions in the skin occurs through epithelial sodium channels (ENaC) located on keratinocytes [20]. The transport of chloride ions is carried out by the cystic fibrosis transmembrane conductance channel (CFTR), chloride channels (ClCs) located on the outer side, and the chloride-sodium-potassium co-transporter located on the opposite side of the skin cells [23]. The transport of sodium and chloride ions in the skin is involved in the perception of stimuli, proper hydration, skin regeneration processes, and interaction with immunocompetent cells in the onset of hypersensitivity and/or allergy reactions [20,21,22,23].

Drugs can change the transport of ions, especially sodium absorption and chloride release, influencing cell functions and the perception of stimuli [18,19,20,21]. Both the level of skin hydration and the hydration of individual cells belong to essential factors affecting the adequate perception of stimuli, including pain stimuli, and the reaction to them. The degree of skin hydration can also affect the production of collagen and elastin, the skin regeneration process, and homeostasis maintenance [24,25,26,27]. The skin with a physiological amount of water is a barrier to microorganisms and external factors and can naturally exfoliate the epidermis [28,29,30,31,32]. The proper state and reactivity of the skin are related to its homeostasis in the transport of water and ions within the cells of the skin tissue [17,23]. Changes in the water transport across skin cell membranes to and from intercellular space result from changes in the transport of ions, mainly sodium and chloride [18,19].

There is no clear evidence of diclofenac’s effect on ion transport processes in the skin. Thus, the study aimed to assess the diclofenac gel’s impact on skin ion transport. Due to the similarity to human skin in terms of similar sensitivity to chemical substances, such as xenobiotics, rabbit skin seems to be a suitable object to assess the effect of various compounds on ion transport [17]. The obtained results can be related to the human skin [17,18,19]. The research model used fragments of rabbit skin with preserved layering and nerve endings. During the study, several electrophysiological parameters, including the transepithelial electrical potential of the skin measured in a stationary state (PD), the maximal and minimal electrical potential of the skin measured during a mechanical or mechanical-chemical stimulation (PDmax and PDmin), as well as transepithelial electrical resistance (R) were assessed.

## 2. Results

After applying diclofenac gel to the skin sample for 15 min, the median R-value was 1416 Ω/cm^2^ at the beginning of the experiment, which was a statistically significant increase over the control (the Mann-Whitney test, *p* = <0.001). The comparison of R measured for the tissue samples treated with diclofenac showed a substantial decrease in resistance at the end of the experiment, i.e., after 15 min of applying a mechanical or mechanical-chemical stimulation (the Wilcoxon test, *p* = <0.001), which was not observed for the control fragments. The R final was also significantly higher for the samples treated with diclofenac compared to the controls (Table 2). Thus, the use of diclofenac gel resulted in increased permeability to ions of the tested skin preparations.

The PD measurement at the beginning of the experiment for the skin samples treated with diclofenac showed a positive potential value of 0.56 mV (median), which was significantly higher compared to the controls (the Mann-Whitney test, *p* = <0.001). Rinsing the tissue samples with stimulating solutions reduced the PD final to values close to the control values (Table 3). Moreover, a comparison of the PD initial and final measurements for the samples treated with diclofenac showed a significant decrease in PD to −0.12 mV (median), which may suggest a short-term effect and the possibility of leaching the drug from the skin specimens.

Each solution used for a 15-s stimulation caused reproducible and measurable changes in the PDmax and PDmin in the tested and control skin fragments (Table 4). The potential recorded during stimulation (PDmax and PDmin) for the controls and the study fragments was always different from the PD measured in stationary conditions, i.e., without stimulation (the Wilcoxon test, Table 5). The measurement of PDmax and PDmin during an RH mechanical stimulation showed similar reactions in the samples treated with diclofenac gel and the controls. Similarly, the use of the sodium ion transport inhibitor (A), the chloride transport inhibitor (B), or both inhibitors together (AB) showed comparable tissue responses. No statistically significant differences existed between the groups for stimulations (Table 5). 

The PDmax and PDmin values measured during the stimulations were subject to analogous changes in the exact directions for both control and diclofenac-treated specimens, regardless of the stimulating solution used. The highest PDmax values of 0.52 mV (median) and the lowest PDmin values (−0.34 mV) were observed for the stimulation with solution A for the tissue samples treated with diclofenac. Lower values were obtained for the controls, but these differences were not statistically significant. Using sodium and/or chloride ion transport inhibitors increased the electronegativity of the measured potential concerning the RH stimulation but did not affect the measured PD in conditions without stimulation for both examined groups. According to these results, it could be concluded that diclofenac did not alter the ability of the skin tissue to open and close ion channels rapidly.

## 3. Discussion

The skin, the body’s largest organ, is the body’s primary barrier protecting against various factors, i.e., UV radiation, mechanical injuries, pathogens, and xenobiotics [17,26,27,28,30,31]. Through to its diversity, the skin is well adapted to perform different functions depending on its location. Among others, it is involved in temperature regulation, water management, the metabolism of vitamin D and lipids, and the perception of various stimuli, including touch and pain [17,30,31]. The proposed experimental model for assessing electrophysiological parameters in the modified Ussing chamber is based on an analysis of full-thickness skin fragments with preserved layered structure and nerve endings [18,19,33]. The Ussing chamber has been frequently used to analyze pathomechanisms of various disorders and diseases associated with the disrupted function of ion transporters and/or ion channels [33,34,35]. The modification of the Ussing chamber involved positioning an analyzed specimen horizontally and its mechanical stimulation by fluids using a nozzle located 3–5 mm above the surface of the examined tissue sample [18,19,33]. This modification extends the use of the Ussing chamber, enabling the study of the impact of mechanical and mechanical-chemical stimulations on the transport of ions in the tested tissue preparations.

The transport of ions and water across cells is essential for maintaining the skin’s continuity as a protective barrier. The ability to respond to stimuli both from the inside of the body and the external environment is closely correlated with changes in the transport of ions, especially sodium and chloride ions [17,22]. It has been proven that the constant transportation of water and sodium, chloride and potassium ions are associated with the sensory system and the initiation of hypersensitivity and/or allergy reactions [3,4,5]. Changes in the directed transport of water and ions can be associated with the formation of micro-deformations and sites with changed water content along with the sensation reactions, e.g., pain or development of pain hypersensitivity [18,19,20,21,23,29]. Drugs, by altering ion transport, especially sodium absorption and chloride release, can interfere with the activity of receptors and transporters located in the cells [18,19,20,21,29].

NSAIDs, including diclofenac, act by inhibiting COX-1 and COX-2, thereby blocking the formation of prostaglandins and thromboxanes in the dermis cells, which are involved in the formation of the inflammatory reaction, fever, and pain transmission [5,36]. The form in which the drug is taken significantly affects the rate and amount of the active substance absorbed into the subcutaneous tissue. Preparations characterized by a relatively fast action are enriched by adding substances that improve skin penetration, which is one of the reasons for using gel preparations [5,16].

Using the Ussing chamber to measure changes in ion transport is a valuable tool for assessing the functioning of ion pumps, channels, and co-transporters and, thus, for determining the response of the tissue to the stimuli applied [18,19,33]. The examined skin fragments were full-thickness, with preserved layering and nerve endings, which allowed the assessment of the cell’s ability to modify ion transport after the incubation with diclofenac, a mechanical stimulation with the iso-osmotic Ringer’s solution or mechanical-chemical stimulations with the blockers of sodium ion transport, namely amiloride, and/or chloride ion transport, namely bumetanide (Table 5).

The high electrical resistance value is a characteristic electrophysiological parameter of the skin due to closely adjacent cells and multilayer structure [33,34,35,37]. In addition, the individual cells that build the skin have a preserved polarity in ion and water transport [33,34,35,37]. Therefore, changes in R depend on changes in the transport of ions, the function of channels and transporters, and the proper tissue structure, especially in appropriate spaces with limited water content [18]. Any decrease in the cell viability or tissue compactness can be observed by analyzing fluctuations in the R values for examined skin fragments [18,19,20,21,33]. In the presented study, all tissue fragments exposed to diclofenac remained alive and entirely reactive, as indicated by the obtained R values (the Mann-Whitney test, Table 2). Interestingly, in the presented study, a significant increase in R was observed after the use of diclofenac compared to the controls, which may be at least partially caused by the effect of the diclofenac gel on the reduction of the amount of water in the intercellular spaces and, consequently, the tighter adhesion of the cells. A similar effect has been observed in the case of skin dryness [33]. However, according to the results of the presented study, this effect of the use of the diclofenac gel seems to be only temporary. It was observed that rinsing the skin surface several times with the stimulating solutions decreased R. Thus, diclofenac penetrating the tissue might be partially washed out by the applied stimuli. 

The R values measured for the samples treated with diclofenac at the end of the experiment were statistically significantly reduced compared to the R values recorded at baseline (the Wilcoxon test, Table 2). The phenomenon of R decrease at the end of the experiment was not observed for the control fragments. This may suggest that diclofenac gel applied to the skin’s outer surface for 15 min could penetrate the cells and/or intercellular spaces and affect individual cell functioning. The decrease in resistance indicates that diclofenac induced an increase in the cell membrane permeability to positive ions, i.e., sodium ions, and modified cellular or intercellular transport. Perez Vallina et al. [36] demonstrated diclofenac’s activation of the sodium-calcium exchanger, which might also occur in the skin cells. However, the change observed in the presented study appears to be transient. Although the R values decreased by the end of the experiment, they did not reach the physiological values recorded for the controls. In addition, by co-analyzing changes in the PD values, it can be concluded that there was a temporary increase in the intracellular transport of sodium ions. It is worth emphasizing that, in the presented study, we demonstrated the immediate effect of diclofenac on changes in the R values measured, which were partially reversible. Undoubtedly, the impact of diclofenac on the skin is more complex than the effect on the COX-2 enzyme activity [4,5,16,37]. The impact of diclofenac on the change in the transport of sodium ions and the possibility of rinsing the drug from the skin *stratum corneum* seems to support the analgesic and anti-inflammatory effects of this drug.

During the experiment, in the case of the controls, we observed a constant transport of ions in the skin. From this observation, it can be concluded that the operation of the sodium-potassium pump and other transporters was not disturbed and did not undergo sudden changes during the entire study. Moreover, the examined tissue fragments corresponded to the applied stimuli with repetitive changes in the measured values of PDmin and PDmax. In contrast, in the case of the skin fragments treated with diclofenac ointment, an increase in PD, which became electropositive, was observed. The significant increase in the measured PD at the beginning of the experiment in the diclofenac-treated samples might result primarily from the intensification of the transport of sodium ions. Additionally, a decrease in the transportation of negative ions, mainly chloride, to the cells might occur (the Mann-Whitney test, Table 3). At the end of the experiment, after diclofenac was washed out of the tissue samples by applying a series of stimulations, the PD values became negative and similar to the results obtained for the controls. Thus, the effect of diclofenac on the intensification of sodium transport was transient. The intensification of sodium transport during the action of diclofenac might be associated with its impact on immunocompetent cells and the initiation of hypersensitivity and/or allergy reactions [18,19,20,21,33].

In the case of every skin sample examined, the application of 15-s mechanical or mechanical-chemical stimuli caused measurable and reproducible changes in the transport of sodium and chloride ions, which differed from the PD values measured in stationary conditions (the Wilcoxon test, Table 5). The applied stimuli induced ion transport changes for the controls and skin samples treated with diclofenac. The measurement of PDmax and PDmin during mechanical and mechanical-chemical stimulations of the study samples showed no differences from the controls (the Mann-Whitney test, Table 4). Physiological changes in the transport of sodium and chloride ions occurring in response to mechanical and mechanical-chemical stimuli are the basis for initiating the stimulus reception reaction [17,18,19,24,33]. In the analyzed system, diclofenac did not seem to affect the perception of skin stimuli. This should be concluded that, despite its analgesic effect, the mechanism of action of diclofenac is not based on a rapid change in cell excitability and stimuli conduction.

## 4. Materials and Methods

### 4.1. Chemicals and Solutions

The following reagents and solutions were used in the experiment:RH—iso-osmotic Ringer solution: K^+^ 4.0 mM; Na^+^ 147.2 mM; Ca^2+^ 2.2 mM; Mg^2+^ 2.6 mM; Cl^−^ 160.8 mM (Avantor Performance Materials, Gliwice, Poland); pH = 7.4, used as a basic solution;B—bumetanide, 3-butylamino-4-phenoxy-5-sulfamoylbenzoic acid, 0.1 mM, 364.42 g/mol (Sigma-Aldrich, St. Louis, MO, USA), used as an inhibitor of transepithelial chloride transport pathways.A—amiloride, 3,5-diamino-6-chloro-2-carboxylic acid, 0.1 mM, 266.09 g/mol (Sigma-Aldrich, St. Louis, MO, USA), used as an inhibitor of transepithelial sodium transport pathways.AB—a solution of amiloride (A, 0.1 mM) and bumetanide (B, 0.1 mM).Diclofenac—a gel containing *diclofenacum natricum*, 10 mg/g (Perrigo, Poland).

### 4.2. Experimental Procedure

The skin samples from 3 New Zealand White rabbits, 2–3 months old, both sexes, weighing 3.5–4.0 kg, were used in the study. The rabbits were killed with carbon dioxide (60% in the air); two methods by a qualified person confirmed the death of the animals. Several tissue and organ samples, including the skin from the abdomen and ears, intestines, trachea, muscles, bladder, heart, kidneys, liver, cartilage and blood, were collected for further study, allowing for the most efficient use of the experimental animals. The presented experiment was carried out on 55 abdomen skin specimens with an area of 1 cm^2^ each. The skin fragments were severed, and the membranous part, muscle, fat, and vessels were discarded. Such prepared specimens were randomly divided into two groups. Next, the outer surface of each study skin specimen (*n* = 31) was smeared with 1 g of diclofenac gel, and then each sample was placed in a petri dish so that the inner skin layer was immersed in RH. The study specimens were then left in a dark place at a constant humidity of 55% and a temperature of 24 °C for 15 min. Control fragments (*n* = 24), not smeared with diclofenac, were immersed in RH for 15 min. Then, each skin fragment was placed in a horizontal position in the modified Ussing chamber, with RH as a fluid filling the chamber. The modification of the Ussing chamber resulted in stimulating the outer layer of the skin with fluid from the peristaltic pump with a flow rate of 0.06 mL/s (1 mL/15 s). Mechanical stimulation of the skin *stratum corneum* was performed using the iso-osmotic RH solution, and mechanical-chemical stimulations were performed using solutions B, A, and AB, respectively. The applied mechanical stimulus imitated the movement of a falling drop of liquid on the surface of the tissue sample. The stimulation nozzle was placed 3’5 mm above the surface of the examined skin specimens. To equalize the pressure and eliminate excess fluid, the Ussing chamber walls were equipped with small holes, allowing excess fluid to flow out. After the stabilization of electrophysiological parameters, the measurement of the parameters and a series of stimulations for all specimens were performed. Each stimulus lasted 15 s, followed by a 2-min break, the measurement of R, a 2-min break, and another stimulation. The experiment lasted 15 min and was performed once for each specimen tested due to possible peripheral damage of the analyzed fragment during a prolonged stay in the apparatus.

### 4.3. Measurement of Electrophysiological Parameters

The electrophysiological parameters were tested in the modified Ussing chamber in stationary conditions and during mechanical or mechanical-chemical stimulation on the skin fragments taken from the outer side of the abdomen of experimental animals. The following parameters were measured during the experiment:-R—transepithelial electrical resistance was recorded while the tissue sample was exposed to a current with a stimulus intensity of ±10 μA; after measuring the voltage change, calculations were made according to Ohm’s law (Ω/cm^2^).-PD—changes in transepithelial electric potential difference measured in stationary conditions, i.e., without a stimulation, recorded continuously (mV).-PDmax and PDmin—minimal and maximal transepithelial electric potential difference measured during a 15-s stimulation (mV).

### 4.4. Data Analysis

Data were recorded in the EVC 4000 experimental protocol (WPI, Worcester, MA, USA). The protocol was connected to the MP150 data acquisition system and transferred to the AcqKnowledge 3.8.1 computer software (Biopac Systems, Inc., Goleta, CA, USA). The results were presented as medians and quartiles. Statistical analysis was performed in Statistica 13.1 (StatSoft, Inc., Krakow, Poland). The non-parametricity of the data distribution was confirmed by the Kolmogorov-Smirnov test, with the Lilliefors correction. The Wilcoxon test was used to compare data from the same incubation conditions with a significance of *p* < 0.05. The Mann-Whitney test was used to detect significant differences (at *p* < 0.05) for different experimental conditions.

## 5. Conclusions

The measurements of electrophysiological parameters in the modified Ussing chamber to measure ion transport in the skin reflects the changes in the microenvironment that occur in and around the skin cells in vivo. Thus, the applied research model enabled the analysis of changes in the transport of ions occurring in the skin after using the gel with diclofenac. The experiments demonstrated that the use of gel with diclofenac caused changes in transepithelial potential difference and in electric resistance. In addition, the potential measured during mechanical stimulation was electropositive. These changes indicate the transient intensification of the sodium ion transport of the fragments treated with diclofenac gel. This process seems to be involved in the mechanism of action of this drug on immunocompetent cells and the initiation of hypersensitivity and/or allergy reactions. Thus, even a short contact of the skin with the diclofenac gel might cause allergic side effects. Nevertheless, further research is needed to explain the role of the transepithelial transport of other ions in the diclofenac gel action on the skin.

## Figures and Tables

**Figure 1 molecules-28-01332-f001:**
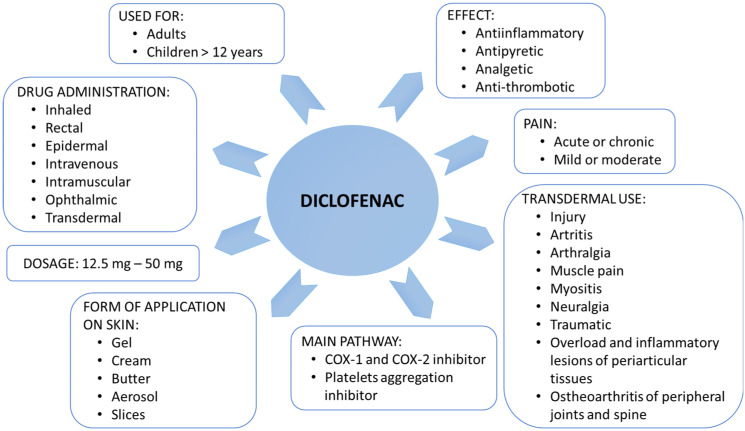
Application and action of diclofenac.

**Table 1 molecules-28-01332-t001:** Incidence of side effects of the application of diclofenac gel to the skin in humans.

1–10/100 Cases
Rash
Erythema
Eczema
Itching
Urticaria
Dermatitis
**1–10/10,000 Cases**
Follicular eruptions
**<1/10,000 Cases**
Edema
Light hypersensitivity

**Table 2 molecules-28-01332-t002:** Transepithelial electric resistance (R) measured at stationary conditions for the skin specimens treated with diclofenac gel and the control skin specimens.

R (Ω/cm^2^)		Control (*n* = 24)	Diclofenac (*n* = 31)	Results of the Mann–Whitney Test (*p*) Control vs. Diclofenac
R initial	median	488	1416	<0.001
	lower quartile	344	835	
	upper quartile	817	1848	
R final	median	498	1338	<0.001
	lower quartile	337	794	
	upper quartile	804	1801	
Results of the Wilcoxon test (*p*) R initial vs. R final		0.24	<0.001	

Abbreviations: R—resistance (Ω/cm^2^), Control—rabbit skin specimens incubated in the Ringer solution, Diclofenac—rabbit skin specimens treated with diclofenac gel for 15 min, *p* < 0.05.

**Table 3 molecules-28-01332-t003:** Transepithelial electric potential (PD) measured in stationary conditions for the skin specimens treated with diclofenac gel and the control skin specimens.

PD (mV)		Control (*n* = 24)	Diclofenac (*n* = 31)	Results of the Mann–Whitney Test (*p*) Control vs. Diclofenac
PD initial	median	0	0.56	<0.001
	lower quartile	−0.12	0	
	upper quartile	0.15	0.51	
PD final	median	0	−0.12	0.25
	lower quartile	0	−0.31	
	upper quartile	0.23	0.19	
Results of the Wilcoxon test (*p*) PD initial vs. PD final		0.44	<0.001	

Abbreviations: PD—transepithelial electric potential measured in stationary conditions (mV), Control—rabbit skin specimens incubated in the Ringer solution, Diclofenac—rabbit skin specimens treated with diclofenac gel for 15 min, *p* < 0.05.

**Table 4 molecules-28-01332-t004:** Maximal (PDmax) and minimal (PDmin) transepithelial electric potential measured during a 15-s mechanical or mechanical-chemical stimulation for the skin specimens treated with diclofenac gel and the control skin specimens.

Stimulating Solution		Control (*n* = 24)		Diclofenac (*n* = 31)		The Mann-Whitney Test (*p*)	
		PDmax (mV)	PDmin (mV)	PDmax (mV)	PDmin (mV)	Control vs. Diclofenac	
						PDmax	PDmin
RH	median	0.15	0	0.37	0	0.16	0.31
	lower quartile	0	−0.18	0.15	−0.21		
	upper quartile	0.43	0	0.67	0.43		
B	median	0.18	−0.12	0.24	−0.21	0.99	0.60
	lower quartile	0.15	−0.21	0	−0.49		
	upper quartile	0.4	0.18	0.67	0		
A	median	0.18	0	0.52	−0.34	0.18	0.39
	lower quartile	0.15	−0.24	0	−0.55		
	upper quartile	0.34	0	0.98	0		
AB	median	0.18	−0.12	0.18	−0.31	0.66	0.35
	lower quartile	0	−0.4	−0.12	−0.64		
	upper quartile	0.52	0	0.88	0		

Abbreviations: PDmax—maximal transepithelial electric potential measured during a 15-s mechanical or mechanical-chemical stimulation (mV), PDmin—minimal transepithelial electric potential measured during a 15-s mechanical or mechanical-chemical stimulation (mV), RH—the iso-osmotic Ringer solution, B—bumetanide (0.1 mM) solution, A—amiloride (0.1 mM) solution, AB—amiloride (0.1 mM) and bumetanide (0.1 mM) solution, Control—rabbit skin specimens incubated in RH, Diclofenac—rabbit skin specimens treated with diclofenac gel for 15 min, *p* < 0.05.

**Table 5 molecules-28-01332-t005:** The Wilcoxon test of maximum (PDmax, mV) and minimum (PDmin, mV) transepithelial electric potential measured during a 15-s mechanical or mechanical-chemical stimulation of the skin specimens treated with diclofenac gel and the control skin specimens.

		*p*-Value	
Stimulating Solution	Parameters Compared	Control (*n* = 24)	Diclofenac (*n* = 31)
RH	PD/PDmax	<0.001	<0.001
	PD/PDmin	<0.001	<0.001
	PDmax/PDmin	<0.001	<0.001
B	PD/PDmax	0.06	<0.001
	PD/PDmin	0.01	<0.001
	PDmax/PDmin	<0.001	<0.001
A	PD/PDmax	<0.001	<0.001
	PD/PDmin	0.01	<0.001
	PDmax/PDmin	<0.001	<0.001
AB	PD/PDmax	0.01	<0.001
	PD/PDmin	0.03	<0.001
	PDmax/PDmin	<0.001	<0.001

Abbreviations: RH—the iso-osmotic Ringer solution, B—bumetanide (0.1 mM) solution, A—amiloride (0.1 mM) solution, AB—amiloride (0.1 mM) and bumetanide (0.1 mM) solution, Control—rabbit skin specimens incubated in the Ringer solution, Diclofenac—rabbit skin specimens treated with diclofenac gel for 15 min, *p* < 0.05.

## Data Availability

Data will be able on request, email: igaholynska@cm.umk.pl.

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
