# Peer review of "The Impact of Diclofenac Gel on Ion Transport in the Rabbit (Oryctolagus cuniculus) Skin: An In Vitro Study"

_molecules, 2023, doi:10.3390/molecules28031332_

Round 1
Reviewer 1 Report
Thank you for the possibility to review the paper titled: The Impact of Diclofenac Gel on the Skin Ion Transport: 2 An In vitro Study
It is well written and designed study. I have few minor comments:
1. The study was conducted on the skin tissue in in vitro model of the study but the tissue was collected from 3 animals which were euthanized for this purpose. I would suggest to correct the title.
2. Materiał and Methods: add information about ethical permissions. Where animals used only in this experiment or this paper is a part of bigger study and the other tissues were also used? If yes add this information to the main text.
3. Remove first sentence from the conclusions - its need to ne supported by relevant citation. Rethink conclusions by interpreting your findings at a higher level of abstraction than discussion.
Author Response
We thank the Reviewer for the careful review of our manuscript and for the positive assessment of our article.
- The study was conducted on the skin tissue in in vitro model of the study but the tissue was collected from 3 animals which were euthanized for this purpose. I would suggest to correct the title.
Ad. 1.
The title has been corrected according to the Reviewer’s suggestion.
- Materiał and Methods: add information about ethical permissions. Where animals used only in this experiment or this paper is a part of bigger study and the other tissues were also used? If yes add this information to the main text.
Ad. 2.
The rabbits were killed to collect numerous tissue and organ samples, including the intestine, trachea, ear skin, heart, kidneys, liver, bladder and skeletal muscles to well use of animals. However, each of the organs was used in separate studies and by separate research teams, which is why this information was not included in the article. However, according to the Reviewer’s recommendation, we have added this information in the Material and methods (subchapter Experimental procedure). We have also added an appropriate statement in the section "Institutional Review Board Statement".
Additionally, we would like to explain that, according to the "DIRECTIVE 2010/63/EU OF THE EUROPEAN PARLIAMENT AND OF THE COUNCIL of 22 September 2010 on the protection of animals used for scientific purposes", the bioethical committee agreement was not required to perform the experiment described in the article because it did not include living animals. The Directive applies to the use of animals in procedures and killing animals solely for the use of their tissues is not such a procedure according to the definition presented in the Directive:
Article 1, no. 2. "This Directive shall apply where animals are used or intended to be used in procedures, or bred specifically so that their organs or tissues may be used for scientific purposes. This Directive shall apply until the animals referred to in the first subparagraph have been killed, rehomed or returned to a suitable habitat or husbandry system."
and Article 3, no. 1 "(…) ‘procedure’ means any use, invasive or non-invasive, of an animal for experimental or other scientific purposes, with known or unknown outcome, or educational purposes, which may cause the animal a level of pain, suffering, distress or lasting harm equivalent to, or higher than, that caused by the introduction of a needle in accordance with good veterinary practice. This includes any course of action intended, or liable, to result in the birth or hatching of an animal or the creation and maintenance of a genetically modified animal line in any such condition, but excludes the killing of animals solely for the use of their organs or tissues (…)"
- Remove first sentence from the conclusions - its need to e supported by relevant citation. Rethink conclusions by interpreting your findings at a higher level of abstraction than discussion.
Ad. 3.
The first sentence has been removed and the conclusions have been reconsidered.

Reviewer 2 Report
I recommend the manuscript titled "The impact of diclofenac gel on the skin ion transport. An in vitro study" to be accepted after minor revision done, therefore I propose:
1) In the introduction, information about the action of diclofenac and about channels involved in the ions’ transport in the skin should be provided.
2) In my opinion, information about COX-3 is redundant.
3) Correct the typo in Figure 1 – it should be spine instead of spin.
Author Response
We thank the Reviewer for the careful review of our manuscript and for the positive assessment of our article.
1) In the introduction, information about the action of diclofenac and about channels involved in the ions’ transport in the skin should be provided.
- 1)
We have added the information about diclofenac action and ion channels.
2) In my opinion, information about COX-3 is redundant.
- 2)
We have removed the information about Cox-3.
3) Correct the typo in Figure 1 – it should be spine instead of spin.
- 3)
The typo in the figure has been corrected.

Reviewer 3 Report
In the manuscript entitled "The Impact of Diclofenac Gel on the Skin Ion Transport: An In vitro Study" written by Dobrzeniecka et al., the authors showed that diclofenac treatment increases the resistance and electric potential of rabbit skin specimens. Although the topic of this study is interesting, this reviewer has a couple of concerns about the manuscript.
Major comments:
In the Materials and Methods section, the authors stated that skin specimens of the diclofenac group were smeared with diclofenac gel and then immersed in Ringer solution while those of the control group were not smeared and then immersed in Ringer solution (Lines 282-287). This reviewer does not think that the control group works as a control because the authors did not smear skin specimens of the control group with the gel not containing diclofenac. Because diclofenac gel contains fast-acting preparations (Lines 56-58), which may affect the results of this study, the authors should prepare the gel not containing diclofenac for the control group and smear the skin specimens with the control gel.
In Tables 2, 3, and 5, the authors show several p-values of the statistical analysis as zero. Is that true?
Author Response
We thank the Reviewer for the careful review of our article.
In the Materials and Methods section, the authors stated that skin specimens of the diclofenac group were smeared with diclofenac gel and then immersed in Ringer solution while those of the control group were not smeared and then immersed in Ringer solution (Lines 282-287). This reviewer does not think that the control group works as a control because the authors did not smear skin specimens of the control group with the gel not containing diclofenac. Because diclofenac gel contains fast-acting preparations (Lines 56-58), which may affect the results of this study, the authors should prepare the gel not containing diclofenac for the control group and smear the skin specimens with the control gel.
Answer:
We would like to thank the Reviewer for this interesting comment. However, we would like to explain why the control group consists of the unsmeared skin samples. Due to the fact that we wanted to reflect natural conditions as close to reality as possible, we used, as the study conditions, the skin samples smeared with the drug, namely the gel with diclofenac, and, as the control conditions, we used the unsmeared skin samples immersed in the isosmotic Ringer's solution, which does not disturb the skin ion transport (the ion transport in the skin immersed in the isosmotic Ringer solution has been described in the in the paper entitled "Analysis of changes in sodium and chloride ion transport in the skin" Scientific Reports 2020, HoÅ‚yÅ„ska-Iwan I., Szewczyk-Golec K.). During the treatment, patients do not apply a pure gel without diclofenac, i.e. without the active substance. Thus, adopting such a model allowed us to check the effect of the entire drug, the main ingredient of which is diclofenac. The producer states that the other ingredients do not affect the skin. They only serve to dissolve the active ingredient and help the better penetration of diclofenac. This is why the assessment of the effect of inactive ingredients of the gel has not been the subject of our study. This is specified in the title of the reviewed paper – “The effect of diclofenac gel on …”, not “the effect of diclofenac on …”.
In Tables 2, 3, and 5, the authors show several p-values of the statistical analysis as zero. Is that true?
Answer
Taking into account the issue mentioned by the Reviewer, we have checked the data comparison results in the program Statistica again. For both, the Wilcoxon and Mann-Whitney tests, the p-value is given as 0. However, p-values are not usually specified as 0, so we have entered values as p<0.001.

Reviewer 4 Report
The authors described the impact of diclofenac gel on skin ion transport using in vitro study. They followed proper protocols to assess the parameters under consideration. The manuscript can be considered for further processing after addressing the following queries.
1) How many measurements are done in each experiment? Please indicate this in the manuscript.
2) The data should be presented as mean ± SD.
3) The manuscript demands moderate English language revision.
Author Response
Thank you for the possibility to review the article.
1) How many measurements are done in each experiment? Please indicate this in the manuscript.
The electrophysiological parameters of each skin sample were measured during 15 minutes in an Ussing chamber in accordance with the experimental procedure described in Material and methods (page 9-10). The experiment was not repeated for any of the tissue samples due to possible peripheral damage of the analyzed fragments during prolonged stay in the apparatus. This information has been added in the manuscript, according to the Reviewer’s advice. However, it should be emphasized that the experiment lasted 15 minutes for each specimen and the measurements were performed after the stabilization of the electrophysiological parameters. Then, the series of stimulations (one stimulation lasted 15 sec, followed by a 2-min break) were done and the parameters were recorded.
2) The data should be presented as mean ± SD.
ad.2)
Due to the fact that the distribution of the analyzed data was nonparametric as it was assessed by the Kolmogorov-Smirnov test with the Lilliefors correction, we presented the results in the form of the median and upper and lower quartiles. According to the statistics science, the mean and standard deviations should be used only when the data show a parametric distribution. Our analysis was consulted with the experienced biostatistician who advised us to use nonparametric analysis for our data.
3) The manuscript demands moderate English language revision.
ad 3)
The manuscript has been corrected by an English professional.

Round 2
Reviewer 3 Report
The authors have addressed all of this reviewer's comments.